# Microfluidic-Assisted Fabrication of Dual-Coated pH-Sensitive Mesoporous Silica Nanoparticles for Protein Delivery

**DOI:** 10.3390/bios12030181

**Published:** 2022-03-18

**Authors:** Berrin Küçüktürkmen, Wali Inam, Fadak Howaili, Mariam Gouda, Neeraj Prabhakar, Hongbo Zhang, Jessica M. Rosenholm

**Affiliations:** 1Pharmaceutical Sciences Laboratory, Faculty of Science and Engineering, Åbo Akademi University, 20500 Turku, Finland; wali.inam@abo.fi (W.I.); fadak.howaili@abo.fi (F.H.); mariam.t.gouda@gmail.com (M.G.); neeraj.prabhakar@abo.fi (N.P.); jessica.rosenholm@abo.fi (J.M.R.); 2Department of Pharmaceutical Technology, Faculty of Pharmacy, Ankara University, Ankara 06560, Turkey; 3Turku Bioscience Center, University of Turku and Åbo Akademi University, 20520 Turku, Finland

**Keywords:** microfluidics, mesoporous silica nanoparticles, protein delivery, polymer coating, pH responsive

## Abstract

Microfluidics has become a popular method for constructing nanosystems in recent years, but it can also be used to coat other materials with polymeric layers. The polymeric coating may serve as a diffusion barrier against hydrophilic compounds, a responsive layer for controlled release, or a functional layer introduced to a nanocomposite for achieving the desired surface chemistry. In this study, mesoporous silica nanoparticles (MSNs) with enlarged pores were synthesized to achieve high protein loading combined with high protein retention within the MSN system with the aid of a microfluidic coating. Thus, MSNs were first coated with a cationic polyelectrolyte, poly (diallyldimethylammonium chloride) (PDDMA), and to potentially further control the protein release, a second coating of a pH-sensitive polymer (spermine-modified acetylated dextran, SpAcDEX) was deposited by a designed microfluidic device. The protective PDDMA layer was first formed under aqueous conditions, whereby the bioactivity of the protein could be maintained. The second coating polymer, SpAcDEX, was preferred to provide pH-sensitive protein release in the intracellular environment. The optimized formulation was effectively taken up by the cells along with the loaded protein cargo. This proof-of-concept study thus demonstrated that the use of microfluidic technologies for the design of protein delivery systems has great potential in terms of creating multicomponent systems and preserving protein stability.

## 1. Introduction

Microfluidics is defined as the “manipulation of fluids in channels tens of micrometers in size” and is one of the cutting-edge technologies that facilitates the preparation of multicomponent systems by processing very low volumes of fluids (i.e., microliters to picolitres) [1]. Microfluidic mixers that usually consist of multiple inlets and a single outlet and that control the flow rate and channel length as well as the reaction between carrier materials and molecules are widely used. Nanoparticles fabricated via this method allow flexible control of size and surface properties since the flow rate, volume, and mixing rate of each fluid in the microfluidic system can be precisely adjusted without contamination problems [2]. The microfluidics technique offers an accessible, robust, and scalable approach [3]. Continuous flow in microfluidic channels ensures the same quality over time for the formulations obtained, eliminating batch-to-batch variability. Moreover, the volume of fluids flowing through the channels is at the nanoliter level, which can significantly reduce the consumption of reagents [4]. As a practical result, microfluidics has greatly accelerated the development and screening of drugs and nanomaterials and has facilitated their transfer to the clinic [5].

To improve the efficacy of the nanoparticulate systems developed for a particular disease, it is necessary to find the most appropriate formulation parameters for each nanoparticle and loaded active substance. These include the selection of nanoparticle-forming materials and solvents, the determination of the appropriate synthesis and drug loading method, the modification of surface coatings and/or functional groups, and other parameters. Formulation optimization is a long process, so it is important to build a platform that can reliably produce a wide variety of nanoparticles in a short time. Microfluidics is an excellent method for preparing nanoparticles via nanoprecipitation. The process involves the controlled incorporation of a polymer solution (organic) into an aqueous medium. The lack of solubility of the polymer in the aqueous solvent leads to the instantaneous formation of the nanoparticles, assisted by a nucleation and growth mechanism [6]. The fate of nanoparticles formed through nanoprecipitation is mainly influenced by the rate of nucleation, which requires supersaturated solute concentrations. To attain homogenous supersaturation, mixing has to be carried out in a quick fashion [7]. In microfluidics, the mixing time between solvent and antisolvent systems could be greatly improved by constructing different microstructures; thus, it is possible to produce nanoparticles with precise control and reproducibility [8,9]. In addition, nanoparticle size could be tuned by varying the flow rates, mixing times, device geometries, the polarity of the solvents, and the concentrations and ratios of the precursors. The addition of reagents during the mixing process and the high production efficiency of the formulations make the microfluidic system even more attractive for the development of nanoparticle formulations.

Microfluidics is also a versatile approach to encapsulate nanoparticles to form structured nanocomposites such as core/shell particles [10,11]. For instance, in one study, a “smart hybrid” nanocomposite was produced by flow-focusing microfluidic nanoprecipitation, consisting of a pH-sensitive polymeric-compound micelle assembled on the surfaces of mesoporous silica nanoparticles (MSNs). Consequently, MSNs showed enhanced plasma stability and were protected from simulated, external physiological conditions (pH 7.4). This smart hybrid inorganic-polymeric system allowed rapid release of the cargo molecule under simulated acidic, tumoral, and intracellular conditions [12]. In another study, a microfluidic nanoprecipitation platform was developed where core/shell nanocomposites could be efficiently synthesized at superhigh speed in a single, continuous process, and the versatility of the platform was demonstrated by producing different core/shell nanocomposites [13].

Protein-based therapies have developed rapidly in recent years to treat various types of cancer and other serious diseases [14]. In comparison to small molecules, proteins have higher binding selectivity and specificity towards target molecules and may thus possess lower toxicity. However, there are several obstacles to the delivery of proteins to specific tissues or cells, including instability during blood circulation, enzyme breakdown, short half-life, immunogenicity, and difficulty crossing cell membranes. Different nanoparticle systems have been created to encapsulate proteins, preserve them from denaturation and degradation, promote tumor-targeted transport, improve transmembrane efficiency, and control protein release/activity in specified areas [14,15]. Here, MSNs are promising candidate carriers for protein delivery due to their ceramic matrix, which provides efficient protection for fragile cargo molecules; their high biocompatibility, and the easy functionalization of their inner and outer surfaces. Their adjustable pore and particle size, as well as their porosity, make MSNs especially suitable for the loading and delivery of proteinous cargos [16]. Large-pore MSNs (LPMSNs) have a larger pore size (6–50 nm) than the standard 3–4 nm but a small particle size, making them ideal for encapsulating biomacromolecules such as (poly)peptides, proteins, and nucleic acids [17]. The rigid inorganic framework of MSNs can effectively protect proteins from denaturation; therefore, when proteins are loaded into the pores of MSNs, their activity and stability are maintained [18]. The pores of MSNs can be sealed with a gatekeeper system provided by stimuli-responsive surface functionalizations which are, for example, pH-sensitive or redox-sensitive [19]. In addition, amorphous silica is classified by the FDA as a "Generally Recognized as Safe" (GRAS) material [20]. Thanks to all these features, MSNs are very advantageous for applications in the field of nanobiotechnology. 

In this study, we aimed to use LPMSNs as a protein delivery system where lysozyme was used as the model protein, and MSNs with large pores were synthesized to provide high lysozyme loading. MSNs were synthesized with a hydrodynamic size of approximately 120 nm, and after the microfluidic surface modifications, the particle size increased by about 40–50 nm. TEM images of MSNs clearly showed that they were uniform in size, well dispersed, and center-radial dendritic mesopore channels could be observed. Microfluidic technology was employed to coat lysozyme-loaded MSNs first with a cationic polyelectrolyte (poly[diallyldimethylammonium chloride], PDDMA) and subsequently with a pH-sensitive polymer (spermine-modified acetylated dextran, SpAcDEX) (Figure 1) [21]. The first PDDMA coating is intended to protect the 3D protein structure of lysozyme and prevent premature release resulting from the open pore structure of the MSNs and the water-soluble characteristics of proteins. SpAcDEX is not coated directly onto the MSNs because it must first be dissolved in a water-miscible solvent such as ethanol, whereby the dispersion of protein-loaded MSNs in ethanol may result in the denaturation of the lysozyme. Concerning this limitation, the MSNs were first coated with the PDDMA polymer, which is highly soluble in water and thus compatible with formulation with proteins, and subsequently coated with SpAcDEX. The second pH-sensitive polymer layer of SpAcDEX was employed to allow intracellular release of lysozyme from MSNs at acidic pH after cellular uptake. The release mechanism is triggered as acetal groups possessed by SpAcDEX are hydrolyzed in an acidic pH, causing dissolution of the polymer. Additionally, SpAcDEX polymer has reactive terminal NH_2_ groups that provide room for attaching additional functional moieties if desired, thus making SpAcDEX very promising for the intracellular delivery of drug-loaded nanoparticles to cancer cells [22]. Lysozyme was continuously released for up to 24 h from SpAcDEX and PDDMA dual-coated MSNs in an acidic environment in vitro. As a result of studies on HeLa cells, it was observed that the cellular uptake of particles was very high after 24 h, and cell viability was found to be higher than 80% after exposure to MSNs at a concentration of 1 µg/mL for 72 h. Taken together, the results show that microfluidic systems can be a promising approach to the modification of nanoparticles for protein delivery.

## 2. Results and Discussion

### 2.1. Characterization of MSNs

The accumulation of nanoparticles in the range of 100–200 nm in tumors by the EPR effect has been shown in many studies [23,24]. In this study, considering the increase in particle size after microfluidic dual coating, we aimed to prepare MSNs of uniform size, centered around 100 nm. They should also have a large pore size to enable the accommodation of protein drugs. Dynamic light scattering (DLS) measurements showed that the observed mean hydrodynamic particle size of the MSNs was 119.3 ± 3.43 nm, with an average PDI of 0.132 ± 0.037, which reflected the uniformity of the synthesized particles. Also, the net surface charge (zeta potential) of these particles was −19.3 ± 0.7 at pH 7.2, which permitted the loading of a cationic protein (IEP > 7.2) without the need for further surface functionalization. TEM images of MSNs clearly showed that they were highly uniform in size, well dispersed, and that the center-radial mesopore channels could be observed (Figure 2). According to the TEM images, it was observed that the size of the MSNs was below 100 nm and the pore size was approximately 10 nm.

### 2.2. Loading Lysozyme into MSNs

Lysozyme was used as a model protein to mimic a cationic protein structure loaded into oppositely charged MSNs. The loading of lysozyme into MSNs was performed by using PBS of pH 7.4, which provided a pH-controlled environment. The zeta potential of lysozyme under the studied conditions was found to be 0.0312 ± 0.160 mV, which is in agreement with previous results even though the high salt content of PBS may have suppressed the absolute value of the zeta potential [25,26]. Given the notion that protein adsorption should be most favorable near the IEP of the protein, these were deemed suitable loading conditions for the lysozyme. The maximal loading capacity of lysozyme thus obtained was as 284 mg/g on average. This loading capacity was obtained by stirring 2 mg/mL lysozyme solution with 1 mg/mL MSN solution for 1 h at room temperature (Appendix A).

According to the 1-h stirring time results, the formulation was prepared again by increasing the stirring time to 24 h with 2 mg/mL lysozyme solution, conditions in which the highest loading capacity was obtained. For this purpose, a 1 mg/mL MSN solution in PBS was mixed with 2 mg/mL lysozyme solution in PBS with a magnetic stirrer for 24 h in a cold room. The loading capacity of lysozyme increased approximately 1.5-fold with this method and was found to be 430 mg/g. 

### 2.3. PDDMA Coating of MSNs

The process of the nanoprecipitation method, according to Mora-Huertas et al., consists of three stages: nucleation, growth, and aggregation. The particle size is determined by the speed of each step, and the driving force behind this phenomenon is supersaturation, which is defined as the ratio of polymer concentration to polymer solubility in the solvent combination [27]. The ideal operating conditions allow for a high nucleation rate, which is heavily reliant on supersaturation, and a slow growth rate. The procedure’s key variables are connected to the organic phase’s addition conditions to the aqueous phase, such as the mixing rate and phase ratio of the organic and aqueous phases. In this study, microfluidic process parameters, such as flow rates, flow rate ratios, polymer concentration, and MSN concentration, were changed during the investigation to find the optimum parameters that produced the best polymer coating (Table 1). After the application of the microfluidic coating method, DLS analysis was performed and if the particle size increased and the PDI value was appropriate, the coating status of the samples was confirmed by TEM analysis.

The particle size at the 2:40 (mL/h) flow rate ratio of inner fluid (PDDMA solution containing MSNs): external fluid(acetone) was found to be close to the MSN size before coating, and the polydispersity index was high (F1, Table 1). Then the PDDMA concentration was increased to 10 mg/mL at 2:40 (mL/h) (F2) and at a 2:20 (mL/h) (F3) flow rate. Although the particle size and polydispersity index in the appropriate range gave preliminary information about the coating, it was thought that the pores of the MSNs might be indistinctly covered when examined by TEM analysis, but the PDDMA layer was not clearly visible (Figure 3A,B). Particle size and PDI value were not within the appropriate range in F4 and F5 formulations (Table 1). The PDDMA concentration was changed to 20 mg/mL and the MSN concentration was changed to 0.5 mg/mL, and coating was attempted with flow rates of 2:40, 2:60, and 2:80 (mL/h). Particle size and PDI value in F6, F7, and F8 formulations were found in the appropriate range where coating could occur. Afterwards, the imaging results were examined with TEM analysis. MSNs were generally uncoated when the flow rate was 2:40 (F6), but it was seen that the pores were less visible and MSNs were interconnected, indicating that MSNs began to be coated with the polymer (Figure 3C). When the flow rate was increased to 2:60 (F7), it was concluded that almost all MSNs were polymer coated and had little aggregation (Figure 3D). It was also observed that MSNs were slightly coated when the flow rate was 2:80 (F8). The particle size and polydispersity index were found to be very high when the PDDMA concentration was 22 mg/mL and above (F9–F13, Table 1). 

The charge reversal indicated by the zeta potential values in Table 1 reflects a change in overall surface characteristics and thus confirms the creation of the PDDMA polyelectrolyte layer, which, as an outer surface coating, is now directing the net surface charge of the particle system [10]. For uncoated MSNs, the zeta potential was −19.3 mV, but after PDDMA coating, it changed to + 14.5 mV under neutral pH conditions (F7). As a result, MSNs were completely coated without aggregation when 20 mg/mL PDDMA and 0.5 mg/mL MSNs were used in inner fluid and when the inner fluid: external fluid flow rate was 2:60 mL/h. Result obtained from DLS showed that particles in F7 formulation exhibited most consistent PDDMA coating around MSN particles. Thus, in the process particle size of bare MSNs increased from 119 nm to 146.5 nm. It is noteworthy that size of particles in F7 formulation is within 150 to 200 nm range which is preferred particle size to prevent extravasation of particles from the circulation via tumor vascularization due to enhanced permeability and retention (EPR) effect [28]. In addition, the PDI value below 0.3 showed that the particles were dispersed in a narrow range and coated homogeneously [29]. Therefore, the F7 formula was chosen for the second coating step.

### 2.4. SpAcDEX Coating of PDDMA@MSNs

The SpAcDEX polymer exhibits a pH response due to the instability of acetal groups in an acidic environment and can be used for the encapsulation of drugs/particles via a nanoprecipitation method due to its amphiphilic property [30]. Microfluidic parameters such as flow rates, SpAcDEX concentration, and PDDMA@MSN concentration have been varied throughout the experimentation to reach the optimum parameters that achieve the best coating results (Table 2).

When the DLS results of the S1–S5 formulations were examined, it was seen that the particle size increased slightly compared to PDDMA@MSN in all of them, and the appropriate PDI values gave preliminary information about the coating. Since SpAcDEX is cationic, it either had no effect on the zeta potential or caused a slight increase. For this reason, the formulations continued to be analyzed by TEM analysis. At the flow rate of 2:40 (mL/h) (SpAcDEX: 0.1% Pluronic F127), MSNs were coated with SpAcDEX, but the coating layer was found to be too thick and some SpAcDEX particles were found to contain more than one MSN (S1, Figure 4A). Therefore, it was decided to lower the SpAcDEX concentration and increase the PDDMA@MSN concentration. When the SpAcDEX concentration decreased to 1 mg/mL and the PDDMA@MSN concentration was doubled, approximately 50% of MSNs were coated at the flow rate of 2:40 (mL/h), but some SpAcDEX particles had holes indicating MSNs entering and exiting (S2, Figure 4B). When the flow rate was changed to 2:20 (S3), the MSNs were found to be coated but aggregated (Figure 4C). Therefore, the PDDMA@MSN concentration was decreased (S4), but the MSNs were not coated. When the SpAcDEX concentration was reduced to 0.5 mg/mL and a 1 mg/mL PDDMA@MSN concentration was used (S5), the particles were found to be monodisperse and totally coated (Figure 4D).

### 2.5. Stability of Lysozyme

The stability of drug-based protein could be affected by an immediate change or adverse condition, leading to degradation or denaturation, and as a result, the dysfunction of the protein [31]. In this experiment, in order to evaluate the survival over time of encapsulated lysozyme in MSNs during the preparation processes, thermal stability was investigated by nano DSF. As shown in Figure 5, the hydrophobic Try 108 located in the core of the lysozyme is exposed to the surface with increasing temperature, resulting in a shift to a longer wavelength and a positive thermal transition signal [32]. The shift in fluorescence emissions of the intrinsic fluorescence of the tryptophan residue of the lysozyme was collected after heating in the range between 330 and 350 nm. The melting points for the following samples were measured (Figure 5). The comparison of the melting points (Tm (°C)) of the samples with the control shows that 99% ethanol and SpAcDEX have adverse effects on the melting point of lysozyme and, as a result, on the stability of lysozyme. A study showed that using ethanol as a co-solvent for spray drying lysozyme, although decreasing the enzymatic activity of the lysozyme to 25%, did not compromise the conformation of the protein [33]. Both SpAcDEX and ethanol are used in the second coating process, so the protein does not come into contact with them, thus justifying the use of the first PDDMA coating. Another study also confirmed our findings, where the structural stability of the lysozyme in contact with water and organic solvent were investigated using CD [34]. The results showed that there was no significant difference between the control lysozyme sample and the sample in the aqueous buffer solution. This result meets our aim because the purpose of the first coating with PDDMA is to reduce the contact of the protein with ethanol. Sonication, a low percentage of acetone, acetate buffer, PBS, and PDDMA have no significant effect on the stability of lysozyme.

### 2.6. Cellular Uptake

To demonstrate the cellular uptake of lysozyme-loaded, dual-coated MSNs in cells, live cell microscopy was performed. After cellular uptake, it is expected that the SpAcDEX polymer first dissolves upon encountering the acidic conditions of the endo/lysosomes, and subsequently, the water-soluble PDDMA polymer is exposed in such a manner that, in turn, leads to its dissolution. For the purpose of studying the intracellular fate of the nanosystem, 10 µg/mL of particles were added to HeLa cells and live-cell microscopy was performed at 24, 48, and 72 h (Figure 6). Dual-coated MSNs were efficiently taken up by the HeLa cells after 24 h of incubation. In addition, it was observed that the fluorescence signal of MSN (red) and the fluorescence of lysozyme (green) were mostly co-localized at different time points, indicating successful intracellular delivery of the protein, which was still retained within the carrier system. The coexistence of red and green fluorescence still after 72 h indicated a very slow release of lysozyme from the MSNs into the cellular environment.

## 3. Materials and Methods

### 3.1. Materials

The materials and accessories used to assemble the microfluidics chip: Glass slide, 75 × 25 mm (Thermo Scientific, Waltham, MA, USA); Outer capillary, OD = 2 mm; ID = 1.56 mm (World Precision Instruments, Inc, Sarasota, FL, USA); Inner capillary, OD = 1.0 mm; ID = 0.58 mm (World Precision Instruments, Inc, Sarasota, FL, USA); syringe-tip, blunt-end needle (Instech Laboratories, Inc., Plymouth Meeting, PA, USA); micromedical tubing ID = 0.034", OD = 0.052" (Scientific Commodities, Inc., Lake Havasu City, AZ, USA); glue, 5 min epoxy (Devcon, Danvers, MA, USA); Puller Model PN-31 (Narishige, Tokyo, Japan); sandpaper, grit = 1200, only a small piece, (Indasa–Rhynowet, Aveiro, Portugal); diamond-tip glass cutter (Harden, Xi’an, China); Meros High Speed Digital Microscope (Dolomite Microfluidics, Royston, UK); syringe pump (PHD 2000, Harvard Apparatus, Holliston, MA, USA). PDDMA solution, acetone, lysozyme from chicken egg white, cetyltrimethylammonium chloride (CTAC), triethanolamine (TEA), cyclohexane tetraethyl orthosilicate (TEOS), (3-Aminopropyl) triethoxysilane (APTES), fluorescein isothiocyanate (FITC), and ammonium nitrate were purchased from Sigma-Aldrich; tetramethylrhodamine (TRITC) was purchased from Fluka, ethanol from Altia Oyj, and acetone from Honeywell. SpAcDEX was prepared by the conjugation of spermine with partially oxidized acetylated dextran [30,35].

### 3.2. Synthesis of SpAcDEX Polymer

The methodology for spermine-modified acetylated dextran (SpAcDEX) synthesis was adopted from the previously done work of Cohen et al [35]. The procedure was carried out in three steps, i.e., dextran partial oxidation, acetylation of partially oxidized dextran, and spermine conjugation. For partially oxidizing dextran, 2 g of dextran was dissolved in 8 mL of water in a round bottom flask. Thereafter, 440 mg of sodium periodate was introduced into the solution and the reaction mixture was allowed to stir for the duration of 5 h. Next, a cellulose dialysis membrane (MWCO, 3500 g/mL) was used to dialyze the solution (distilled water was changed four to five times). The dialyzed solution was freeze-dried to obtain partially oxidized dextran (PO AcDEX). 

To perform acetylation, 1.324 g of PO AcDEX was dissolved in 13.34 mL of DMSO and reaction was proceeded with the addition of pyridinium p-toulenesulfonate (2.63 mg) and 2-methoxypropene (4.5 mL) in the reaction mixture. Reaction was interrupted after 3 h of stirring with addition of 1.3 mL triethylamine (TEA). Solution was precipitated with distilled water (pH 8) and the subsequent pellet collected after centrifugation was washed and freeze-dried.

Product obtained as a white powder (1740 mg) was reacted with spermine (3480 mg) while both ingredients were dissolved in DMSO (14 mL). The reaction mixer was continuously stirred and heated at 50 °C for 18 h. Then, reduction was performed by adding NaBH4 (3480 mg) into the reaction mixture. Afterwards, the product present in the solution was precipitated with distilled water (pH 8). Precipitates were collected as pellets after centrifugation and washed twice with distilled water (pH 8). The final product (SpAcDEX) was obtained as white powder after freeze-drying the solution.

### 3.3. Synthesis of Non-Labelled and Tetramethylrhodamine (TRITC)-Labelled Mesoporous Silica Nanoparticles

The synthesis of MSNs was performed as follows: 36 mL of milliQ-water was added to the 100 mL flask, and 24 mL of CTAC and 160 µL of TEA were also added. The flask was closed with a glass lid, placed in a paraffin bath, and incubated with stirring at 60 °C for 1 h. Stirring was achieved using a Teflon-coated stirring bar and the stirring rate was adjusted to be ~150 rpm. Meanwhile, 20% TEOS-cyclohexane solution was prepared. After a 1-h incubation period, 20% TEOS solution was added dropwise to the mixture and stirred at 60 °C overnight. The particles were collected by centrifugation at 13,000 rpm at 18 °C for 20 min. Subsequently, the supernatant was discarded, and the precipitate was redispersed by vortexing and sonication in ethanol, and in the end, the precipitate was washed two times by centrifugation.

The template removal of synthesized MSNs was achieved by an efficient ion-exchange method, where they were extracted using the extraction solution (0.6 wt% ammonium nitrate ethanol solution). Extraction solution was added to the precipitate and dispersed by vortexing and sonication and then stirred at 60 °C for 6 h with a magnetic stirrer. At the end of the time, it was centrifuged at 13,000 rpm and 18 °C for 20 min. Hence, washing was carried out twice with the extraction solution. The final precipitate was washed again with ethanol and then extracted particles were preserved in ethanol for further use, where they can remain stable for several months.

For the preparation of TRITC-labeled MSNs, firstly 1.5 mg/mL TRITC–ethanol solution was prepared and allowed to stand in an ultrasonic bath for 2–3 min and afterwards vortexed for 2–3 min to dissolve the TRITC completely, and was then stirred magnetically under vacuum (dye-conjugation solution). After the formation of air bubbles, 0.02 mL APTES was added with the syringe and stirred under vacuum for 2 h. Then, the dye-conjugation solution was added after adding 20% TEOS-cyclohexane solution during the synthesis.

### 3.4. Characterization of Synthesized MSNs

To qualify them for further use, the synthesized MSNs were characterized in terms of hydrodynamic size, net surface charge (zeta potential), and morphology. Hydrodynamic size and zeta potential measurements were performed through DLS and electrokinetic measurements using a Zetasizer NanoZS (Malvern Instrument Ltd., Worcestershire, UK) instrument. For size determination, MSNs were sonicated and dispersed in Millipore water right before placement in a disposable polystyrene cuvette (STARSTEDT AG & Co., Nümbrecht, Germany). The zeta potential was measured by dispersing the MSNs in HEPES buffer (pH 7.2, 25 mM) and loading them in a disposable, folded capillary cell (DTS1070, Malvern, Worcestershire, UK). 

Transmission electron microscopy (TEM) was used (TEM; JEM-1400 Plus Electron Microscope, JEOL, Musashino, Akishma, Tokyo, Japan) for size and morphological analysis of the MSNs. The TEM samples were prepared by sonication and dispersion of the bare MSNs, and the coated MSNs were prepared with PDDMA in ethanol, and with water for the MSNs after the second coating with SpAcDEX. Nanoparticles in the quantity of 10 μL were deposited onto carbon-coated copper grids (Ted Pella Inc., Redding, CA, USA) and later were allowed to air-dry overnight before imaging.

### 3.5. FITC-Lysozyme Bioconjugation

The lysozyme solution was prepared in 0.1 M sodium carbonate (pH 9) at a concentration of 2 mg/mL. In a darkened environment, a certain amount of FITC was dissolved in DMSO, yielding the concentration of 1 mg/mL. Then, 50 μL of the FITC solution was slowly added to each ml of the protein solution and the mixture was stirred gently for 20 h at 4 °C. The mixture was then dialyzed against distilled water for the removal of unconjugated FITC using a dialysis bag of 3500 MWCO (molecular weight cut-off) for two days, and the water was changed each day. The resulting FITC–lysozyme solution was freeze-dried and stored at 4 °C.

### 3.6. Lysozyme Loading

A lysozyme stock solution of 2 mg/mL was prepared in phosphate buffer saline (PBS, pH 7.4) and several dilutions were prepared (0.25, 0.5, 1 mg/mL, and 2 mg/mL) to observe the lysozyme loading efficiency and capacity. A stock of MSNs (1 mg/mL) was sonicated and dispersed in PBS. MSN solution and lysozyme solution were mixed at 1:1 *v*/*v* ratio for 1 h at room temperature with an overhead stirrer or for 24 h in a cold room with magnetic stirrer. Next, the mixture was centrifuged at 13,000 rpm for 5 min to collect the MSNs. The excess, non-loaded lysozyme was collected and assayed using a UV-Vis Spectrophotometer (NanoDrop 2000c, Thermo Fisher Scientific, Wilmington, NC, USA) at a wavelength of 280 nm. The loading efficiency % (LE%) and loading capacity (LC, mg/g) were determined based on the initial and final protein concentrations, according to Equations (1) and (2), using the standard calibration curve.
𝐿𝐸 % = (total drug added − free non-entrapped drug)/total drug added × 100(1)
𝐿𝐶 (𝑚𝑔/𝑔) = (total drug added − free non-entrapped drug)/𝐴𝑚𝑜𝑢𝑛𝑡 𝑜𝑓 𝑀𝑆𝑁s(2)

### 3.7. Fabrication of the Microfluidic Flow-Focusing Glass-Capillary Chip

The microfluidic flow-focusing glass-capillary chip was assembled by the method of Ma et al. (2019), with some modifications [36]. The plastic parts of two syringe tips were heated and softened, and the metal parts were removed. These were used as the input and output of the microfluidics system. At the bottom of the plastic part of another syringe tip, the outer capillary tube was heated and melted with it, forming two opposing V-shaped holes to cover the size of the outer capillary vessel, thereby allowing it to rest on the capillaries. 

In order to taper the end of the inner capillary tube, the glass tube was placed on the puller, fixed with screws, and heated through the middle of the glass tube. After the glass tube was separated, the tip was sharpened with sandpaper until a suitable tip size of approximately 100 nm was obtained (equal to 1 cm on the microscope screen). Compressed air was used to remove the glass particles from the outside of the created tip.

The outer capillary tube was cut into two short pieces and one longer piece with a glass cutter. One small piece was used as a bridge between the inlet of the microfluidics system and the inner capillary part, and the other short piece was used to create the outer phase inlet and hold the longer part.

All parts created were aligned as shown in Figure 7, and their positions were adjusted. Glue was prepared by mixing together hardener and resin and was applied to the inlet and outlet parts first. The plastic part of the syringe tip to which the outer phase would be attached was then fixed with glue. The prepared microchip was allowed to solidify for 1 night without moving. In this way, a microfluidic flow-oriented glass-capillary device with 2 inlets and 1 outlet was designed (Figure 7) [36].

### 3.8. PDDMA Coating

MSNs were coated with PDDMA via a microfluidics-based nanoprecipitation approach [30]. Firstly, PDDMA polymer solutions were prepared in Millipore water at concentrations of 5 mg/mL, 10 mg/mL, 15 mg/mL, 20 mg/mL, 25 mg/mL, and 30 mg/mL. Lysozyme-loaded MSNs were dispersed in water (0.25 mg/mL or 0.5 mg/mL) and then mixed with the polymer solution, yielding a certain concentration ratio of polymer to MSN. In the next step, the MSN–polymer nanocomposites were prepared using the microfluidic flow-focusing glass-capillary device. The latter suspension served as the inner fluid, flowing through the first inlet, while pure acetone served as the counter solvent, flowing through the second inlet as the outer fluid. These two miscible liquids were separately loaded into plastic syringes, where polyethylene tubes were attached to them. Through these tubes, the liquids flowed into the microfluidics device at constant flow rates. The flow rate of the liquids was controlled by the aid of pumps (PHD 2000, Harvard Apparatus, Holliston, MA, USA). The flow pattern was monitored under the microscope (Dolomite Microfluidics, Royston, UK) using a high-speed digital camera. 

Microfluidic parameters such as flow rates, flow rate ratios, polymer concentration, and MSN concentration were varied throughout the experiments to reach the optimum parameters that achieve the best coating results (Table 1). Obtained coated nanoparticles were retrieved by centrifugation for 5 min at 13,000 rpm. Then, the precipitate was dispersed in ethanol to be ready for the next coating.

### 3.9. SpAcDEX Coating

PDDMA@MSNs were coated with SpAcDEX via a microfluidics-based nanoprecipitation approach [30]. Firstly, SpAcDEX solutions were prepared in ethanol at concentrations of 1 mg/mL and 2 mg/mL. PDDMA-coated MSNs (PDDMA@MSN) were dispersed in ethanol and then mixed with the SpAcDEX solution to yield a certain concentration ratio of SpAcDEX to PDDMA@MSN. The coating process was achieved in the same way mentioned in the previous section, through nanoprecipitation using the same microfluidic glass-capillary device. However, this time the drug-loaded PDDMA@MSN, dispersed in ethanol, served as the inner fluid flowing through the first inlet, while the Millipore water containing 0.1% Pluronic F127 (pH 7.6) flowed through the second inlet, serving as the precipitating solvent. Different parameters were monitored to yield the optimum coating results. Obtained coated nanoparticles were retrieved by centrifugation for 10 min at 13,000 rpm.

### 3.10. Stability of Lysozyme

The stability of lysozyme was confirmed by nano differential scanning fluorimetry (Nano DSF) (NanoTemper Prometheus NT 4.8, NanoTemper Technologies GmbH, München, Germany), which is based on the intrinsic fluorescence of tryptophan residue at 330 and 350 nm, located in the hydrophobic core of lysozyme. For this purpose, we investigated the stability of lysozyme by a change in melting point toward different concentrations of processing materials, e.g., sonication, acetone, acetate, PBS, PDDMA, and SpAcDEX. All samples were centrifuged for 5 min, at 13,000 rpm at 4 °C and placed in high-sensitivity glass capillaries (Cat#PR-C006, NanoTemper Technologies, GmbH, München, Germany). The measurement temperature was set in the range (0–95 °C), and the heating rate was (2 °C/min). The result is reported as the melting temperature (Tm), the temperature where half of the protein is unfolded. By employing the Therm Control Software V 2.1.5, the fluorescence intensity ratio (F350/F330) was plotted against different temperatures, and the inflection point (IP350/330) of the transition was calculated from the maximum of the first derivative of each data point.

### 3.11. Cellular Uptake 

HeLa cells (50 k) were plated in a 96-well plate. HeLa cells were maintained in DMEM (10% fetal calf serum (FCS), 1% amino acids, and 1% penicillin–streptomycin). SpAcDEX coated PDDMA@MSNs (TRITC-labelled) loaded with lysozyme (FITC-labelled) were incubated with HeLa cells to evaluate the cellular uptake. Then, 10 ug/mL of particles were added into each well. Cell samples were prepared with complementary control cells. The live-cell microscopy setup consisted of a Leica TCS SP5 (Leica microsystems, Wetzlar, Germany) confocal microscope, PMT, and 100× oil objective. FITC-labeled lysozymes were excited with a 488 nm argon laser and emission was collected by green channel (510–550 nm). The TRITC-labeled PDDMA@MSNs were excited by a 561 nm laser and emissions were collected (575–610 nm). The cells were maintained at 37 °C and 5% CO_2_.

## 4. Conclusions

MSNs are one of the most promising carrier systems in the delivery of protein-based drugs. Since the solid inorganic framework can effectively protect proteins from denaturation, the activity and stability of proteins can be maintained when loaded into the pores of the MSNs. Compared to conventional MSNs, large-pore MSNs have desirable properties for the encapsulation of bio macromolecules such as peptides and proteins. In this context, the purpose of this study was to develop an MSN-based carrier that can be loaded with protein drugs, maintain their stability, and increase their entry into cancer cells. To further increase protein stability, prevent premature release, and increase the uptake into cancer cells, a polymeric coating and a pH-sensitive coating were deposited onto the particles. The coating processes were carried out through a microfluidic capillary device that provided gentle, fast mixing and minimized contact with organic solvents, thereby preserving the activity of the protein.

In this research, large-pore mesoporous silica particles were synthesized and used as protein carriers. They were found promising as a protein delivery system with high loading (430 mg/g) of the model protein drug lysozyme. Microfluidics-assisted polymer encapsulation showed promising, robust results using water-soluble PDDMA polymer. The MSNs coated with PDDMA were then successfully coated with an additional SpAcDEX layer using a microfluidics method for introducing pH-sensitive release ability. Cellular uptake studies have shown that engineered particles can be effectively taken up by HeLa cells after 24 h of incubation. In conclusion, this proof-of-concept study shows that microfluidic technologies have great potential in paving the way for more efficient protein-based drug delivery systems.

## Figures and Tables

**Figure 1 biosensors-12-00181-f001:**
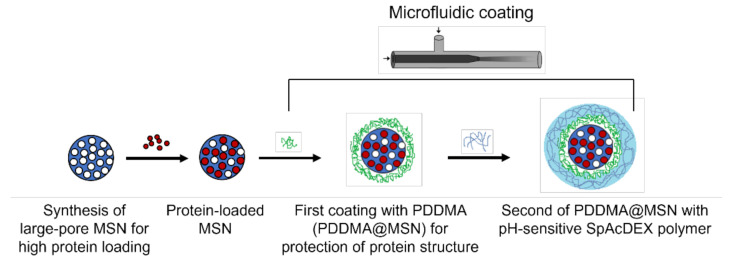
Process scheme applied for microfluidic dual coating of protein-loaded MSNs.

**Figure 2 biosensors-12-00181-f002:**
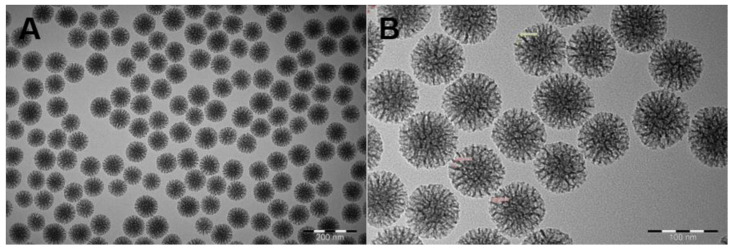
TEM images of MSNs. Images show size and structure uniformity and readily dispersed particles. Scale bars: (**A**) 200 nm, (**B**) 100 nm.

**Figure 3 biosensors-12-00181-f003:**
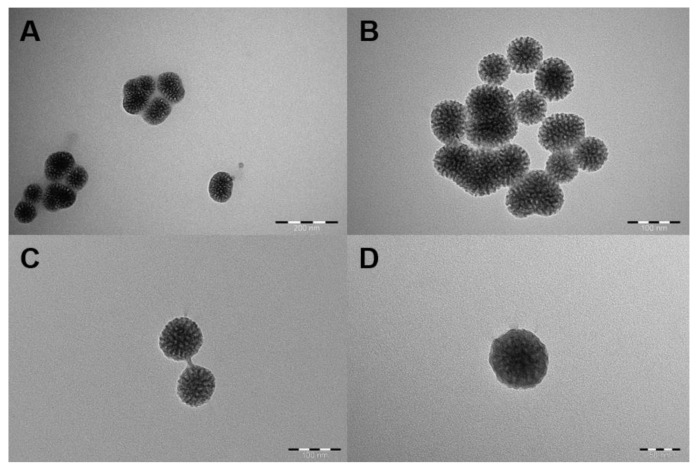
TEM images representing PDDMA polymer-encapsulation results. A: F2, B: F3, C: F6, and D: F7. Scale bars: (**A**) 200 nm, (**B**) and (**C**) 100 nm, and (**D**) 50 nm.

**Figure 4 biosensors-12-00181-f004:**
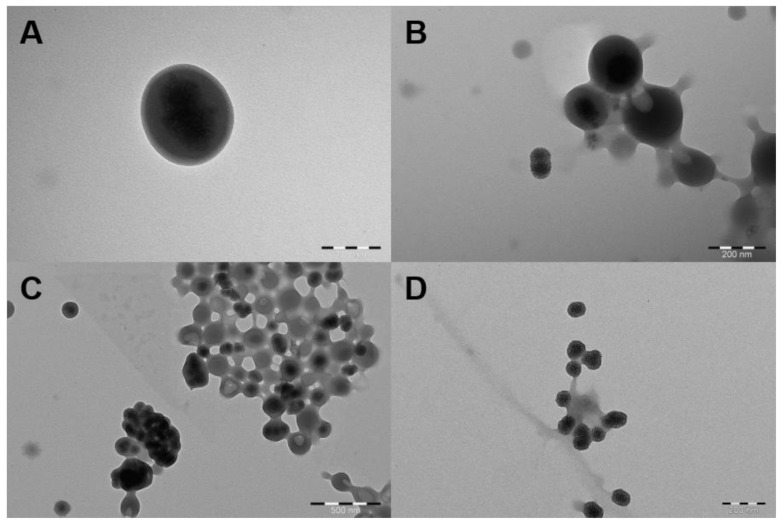
TEM images representing SpAcDEX polymer encapsulation results. A: S1, B: S2, C: S3, and D: S5. Scale bars: (**A**) 100 nm, (**B**) 200 nm and (**C**) 500 nm, (**D**) 200 nm.

**Figure 5 biosensors-12-00181-f005:**
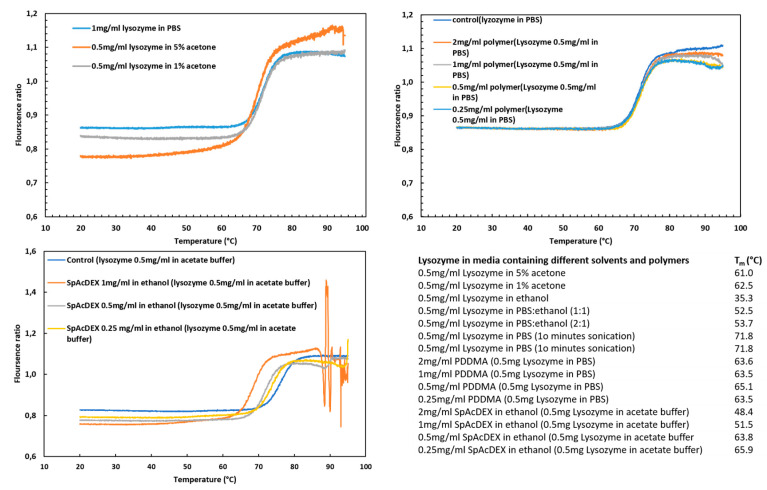
The effect of different solvent- and polymer-containing environments on lysozyme stability.

**Figure 6 biosensors-12-00181-f006:**
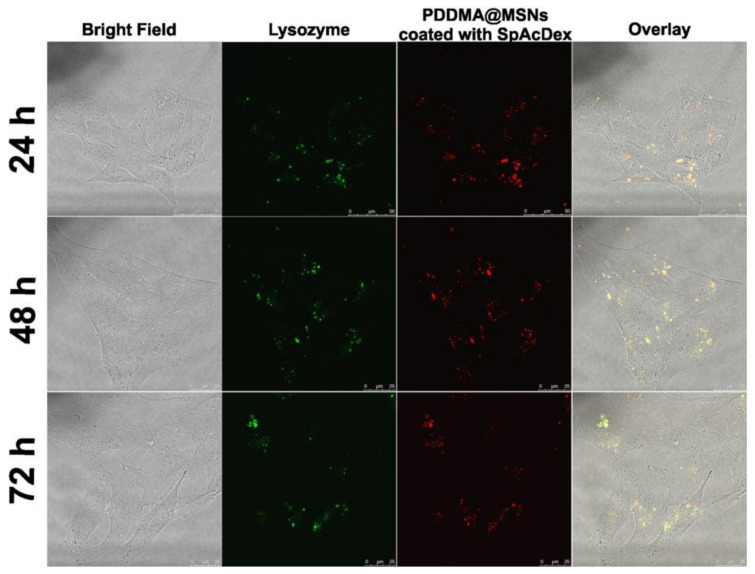
The cellular uptake of lysozyme-loaded (green) SpAcDEX-coated PDDMA@MSNs (red) incubated with HeLa cells at 24 h, 48 h, and 72 h.

**Figure 7 biosensors-12-00181-f007:**
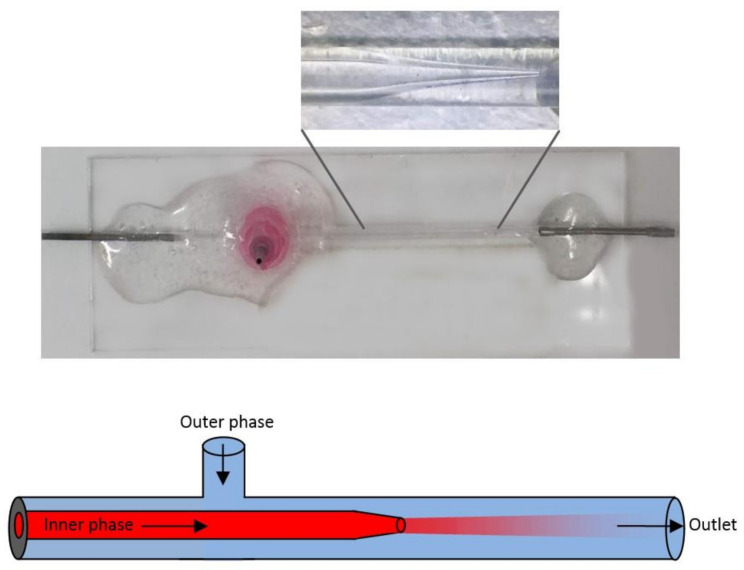
Designed microfluidic chip.

**Table 1 biosensors-12-00181-t001:** Microfluidics parameters for PDDMA coating and characterization results with DLS.

Code	PDDMA (mg/mL)	Flow Rate (mL/h) (PDDMA: Acetone)	MSN (mg/mL)	Size (nm)	PDI	Zeta Potential (mV)
F1	5	2:40	0.25	96.16 ± 4.09	0.401 ± 0.064	−2.02 ± 0.77
F2	10	2:40	0.25	146.5 ± 4.17	0.197 ± 0.025	17.0 ± 0.90
F3	10	2:20	0.25	140.1 ± 3.60	0.385 ± 0.021	9.49 ± 0.90
F4	15	2:40	0.25	105.2 ± 9.45	0.466 ± 0.049	11.1 ± 1.17
F5	20	2:20	0.5	792.5 ± 152.0	0.436 ± 0.238	17.7 ± 0.43
F6	20	2:40	0.5	276.7 ± 2.85	0.151 ± 0.054	33.5 ± 1.30
F7	20	2:60	0.5	146.5 ± 2.12	0.262 ± 0.008	14.5 ± 0.32
F8	20	2:80	0.5	218.1 ± 1.50	0.187 ± 0.012	24.9 ± 0.75
F9	22	2:20	0.5	463.5 ± 59.62	0.384 ± 0.539	22.9 ± 0.43
F10	22	2:40	0.5	396.9 ± 22.16	0.521 ± 0.268	7.63 ± 1.81
F11	22	2:60	0.5	282.2 ± 17.16	0.698 ± 0.365	24.9 ± 0.50
F12	25	2:60	0.5	570.2 ± 108.0	0.827 ± 0.300	14.8 ± 0.60
F13	30	2:60	0.5	517.4 ± 41.25	0.743 ± 0.371	16.5 ± 2.21

**Table 2 biosensors-12-00181-t002:** Microfluidic parameters for SpAcDEX coating and the resulting characterization results.

	SpAcDEX (mg/mL)	Flow Rate (mL/h) (SpAcDEX: 0.1% Pluronic F127)	PDDMA@MSN (mg/mL)	Size (nm)	PDI	Zeta Potential (mV)
S1	2	2:40	2	225.2 ± 2.90	0.316 ± 0.032	26.4 ± 1.11
S2	1	2:40	2	166.2 ± 2.40	0.204 ± 0.014	14.5 ± 0.90
S3	1	2:20	2	173.4 ± 3.77	0.225 ± 0.006	16.5 ± 1.61
S4	1	2:20	1	168.3 ± 3.69	0.184 ± 0.012	20.1 ± 1.33
S5	0.5	2:20	1	166.3 ± 14.06	0.226 ± 0.026	14.1 ± 0.43

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
