# Peer review of "Microfluidic-Assisted Fabrication of Dual-Coated pH-Sensitive Mesoporous Silica Nanoparticles for Protein Delivery"

_biosensors, 2022, doi:10.3390/bios12030181_

Round 1
Reviewer 1 Report
The manuscript, Microfluidic assisted fabrication of dual-coated pH-sensitive mesoporous silica nanoparticles for protein delivery, should be revised to address the following issues:
The introduction part is too long. It should be shortened.
Lines 57, 58, page 2 “…… incorporation of a polymer solution (organic) into an aqueous solvent (water) must be replace with ……. incorporation of a polymer solution (organic) into an aqueous medium.
The explanation of the purpose of the work can be summarize for an easier understanding in a scheme, particularly that the experimental part is after results and discussion part.
Figure 2 can be moved on Supplementary Material. The data from the figure should be corrected by replacing coma with dot.
Lysozyme release profiles are shown in Figure 5, not Figure 6 (page 8, line 279). For the lysozyme release profiles should provide the error bar, especially since the experimental points are quite widespread.
For the release profile some explanations must be provided.
In the figure 7 for the absorbance value replace coma with dots.
Author Response
- The introduction part is too long. It should be shortened.
Response: In line with the author's suggestion, we removed following sentences from introduction part because similar expressions were mentioned at the top of the relevant paragraph. ‘The microfluidic device integrated with a multi-input micro-mixer allows programmable mixing of a wide variety of precursors prior to nanoprecipitation. The composition of the nanoparticles can be easily adjusted by altering the flow rate of each precursor. Therefore, fully integrated microfluidic devices are suitable for the short-term evaluation of a large number of formulations in the development of nanoparticle formulations [5]. Valencia et al. (2013) used an integrated microfluidic platform to develop 45 different nanoparticle formulations with dissimilar characteristics by combining more than 15 types of nanoparticle precursors in various ratios [10]. Data from 648 different nanoparticles were collected in just 2.5 hours by automated sampling, dilution, measurement and mixing of precursors with a microfluidic device designed in a previous study [11].’
- Lines 57, 58, page 2 “…… incorporation of a polymer solution (organic) into an aqueous solvent (water) must be replace with ……. incorporation of a polymer solution (organic) into an aqueous medium.
Response: The sentence has been corrected as suggested by the reviewer.
- The explanation of the purpose of the work can be summarize for an easier understanding in a scheme, particularly that the experimental part is after results and discussion part.
Response: The process scheme has been added to the manuscript as Figure 1 to summarize the purpose of the study.
- Figure 2 can be moved on Supplementary Material. The data from the figure should be corrected by replacing coma with dot.
Response: 4. According to the reviewer’s suggestion, Figure 2 has now been moved to Supplementary Material and the data from the figure corrected by replacing coma with dot.
- Lysozyme release profiles are shown in Figure 5, not Figure 6 (page 8, line 279). For the lysozyme release profiles should provide the error bar, especially since the experimental points are quite widespread. For the release profile some explanations must be provided.
Response: We further compared the in vitro release profiles using the f2 similarity factor to see if there was a significant difference between the profiles and evaluated them in terms of compliance with the release kinetics. The in vitro release test could not be replicated, however, since our aim in this article is proof-of-concept rather than application so this experiment should be viewed as a demontration only; and therefore, we also moved the in vitro release part to the supplementary part. The number of the relevant figure is also corrected accordingly in the supplementary file.
- In the figure 7 for the absorbance value replace coma with dots.
Response: Figure 7 is Figure 6 in the revised manuscript and has been corrected as suggested by the author.
Reviewer 2 Report
SIGNIFICANCE OF THE WORK:
In this paper mesoporous silica nanoparticles for protein delivery by using microfluidic techniques have been prepared. The particles were coated with a doble polymeric layer. The design of such coating endows the system with pH sensitivity.
METHODOLOGY:
The prepared materials have been thoroughly characterized by typical methods.
TEXT:
The manuscript is well-written, and the figures are well crafted and informative.
NOVELTY
To the best of my knowledge the system reported is new.
COMMENTS:
Although the design and synthesis of the system is quite interesting, this work present two main problems that should be addressed before publication:
- Figure 5 shows the release of lysozyme from the particles. A statistic analysis should be carried out to determine whether the values obtained are significantly different. From inspection of the curves, it is not clear to me. Take as example lines red and green. They are almost coincident.
- In my opinion, this work is a proof-of-concept rather than a complete report. The biological characterization is not relevant since the coating is extremely cationic which makes the particles toxic. In fact, the authors recognize that the present system is incomplete: “Here we note that the SpAcDEX coating is not meant to be a coating for final use, but is mainly used for its ample possibilities for further functionalization given its abundance of amine groups.” I would recommend completing this part (conjugation with some ligand, for instance).
Author Response
- Figure 5 shows the release of lysozyme from the particles. A statistic analysis should be carried out to determine whether the values obtained are significantly different. From inspection of the curves, it is not clear to me. Take as example lines red and green. They are almost coincident.
Response: We further compared the in vitro release profiles using the f2 similarity factor to see if there was a significant difference between the profiles and evaluated them in terms of compliance with the release kinetics. However, since our aim in this article is proof-of-concept rather than application, we have now moved the in vitro release part to the supplementary part so as not to overemphasize the significance of this demonstration. We have thus now also highlighted the proof-of-concept aspect in both the Abstract and Conclusions to maket his clear also for the readers.
- In my opinion, this work is a proof-of-concept rather than a complete report. The biological characterization is not relevant since the coating is extremely cationic which makes the particles toxic. In fact, the authors recognize that the present system is incomplete: “Here we note that the SpAcDEX coating is not meant to be a coating for final use, but is mainly used for its ample possibilities for further functionalization given its abundance of amine groups.” I would recommend completing this part (conjugation with some ligand, for instance).
Response: We agree with the reviewer's opinion. Our aim in this article is indeed to show a proof- of-concept rather than application. This was more or less finite work done 2 years ago that was abruptly ended by global lockdown, whereby the first author whose main project this was, had to return back home before the time of the research visit was up. Most results span from this time, so unfortunately it is not possible for us to renew experiments that would require doing everything from the beginning due to lack of resources. As suggested by the reviewer, we see it as an important study that will continue in this direction and form the basis for future studies.
What we meant to say with the above-cited sentence was also not that SpAcDEX would not be possible to use at all, but once the high cationic charge is suppressed by further funcitonlaization (e.g. conjugation with some ligands, as also suggested by the reviewer) it should be more cytocompatible. We have observed this phenomenon multiple times in the past decade when working with PEI as a coating, which in itself is known to be extremely cytotoxic. Once used as coating and with ligands, this effect stemming from the high cationinc charge density disappears. We have now clarified this sentence so as to avoid any misunderstanding in this regard and also moved the cytotoxicity results to the Supplemetary Information since it as such does not add to the main story, but rather serves as a basic characterization method. We have also added another reference where we have already used SpAcDEX as a design component together with cells, to emphasize that SpAcDEX indeed still is possible to use in such designs.
Reviewer 3 Report
The manuscript presents a very complex multidisciplinary study about MSN-based carrier that were loaded with protein drugs, maintain their stability and increase their entry into cancer cells.
The authors stated that the fast mixing and minimize contact with organic solvents, and thereby preserving the activity of protein drugs has been accomplished. Still, the lysozyme enzymatic activity assay would have been good to be evaluated, too.
In the following text fragment:
“Subsequently, the supernatant was discarded and the cake was redispersed by vortexing, and sonication in ethanol, in the end the cake was washed two times by centrifugation”.
The cake? Please explain or reformulate. Also, in raw 425.
Please explain why was need the dialyze step for the preparation of the lysozyme solution. Not all the readers are familiar with this part, since the study is multidisciplinary.
Concerning the drug release study, please add the number of replicates, the statistics too (standard deviation, etc.). Are any kinetic models suitable that can be apply for this kind of formulation that you have obtained?
Concerning the calibration curves that you used for calculation, both the one in PBS and the one in acetate buffer too, it will be good to add them. Please add them into the Supplementary Material, if you consider that it shall not overwhelmed the manuscript, because I thing that the readers will be curious to see all these details.
Author Response
- In the following text fragment: “Subsequently, the supernatant was discarded and the cake was redispersed by vortexing, and sonication in ethanol, in the end the cake was washed two times by centrifugation”.The cake? Please explain or reformulate. Also, in raw 425.
Responce:
Thank you for this correction for the proper spelling of the word in a scientific article. The entire article has been revised and the word 'cake' has been changed to 'precipitate'.
- Please explain why was need the dialyze step for the preparation of the lysozyme solution. Not all the readers are familiar with this part, since the study is multidisciplinary.
Responce:
Dialyze step was used to remove unconjugated FITC after FITC-lysozyme conjugation. Thank you for pointing out this omission; the purpose of the dialysis process has also now been added to the relevant place in the manuscript.
- Concerning the drug release study, please add the number of replicates, the statistics too (standard deviation, etc.). Are any kinetic models suitable that can be apply for this kind of formulation that you have obtained?
Responce:
The in vitro release test could not be replicated due to the reason mentioned above (the work was never possible to complete in full due to global lockdown), but to shed some light on the matter, we have now compared the in vitro release profiles using the f2 similarity factor to see if there was a significant difference between the profiles and evaluated them in terms of compliance with the release kinetics (Supplementary Information). However, since our aim in this article is proof of concept rather than application, we moved the in vitro release part to the supplementary part together with the calibration curves so as not to overemphasize the significance of this demonstration. We have further now also highlighted the proof-of-concept aspect in both the Abstract and Conclusions to maket his clear also for the readers.
- Concerning the calibration curves that you used for calculation, both the one in PBS and the one in acetate buffer too, it will be good to add them. Please add them into the Supplementary Material, if you consider that it shall not overwhelmed the manuscript, because I thing that the readers will be curious to see all these details.
Responce:
Calibration curves of lysozyme in PBS and acetate buffer are added to the supplementary file.
Round 2
Reviewer 1 Report
The manuscript can be published in this form. The authors made the required modifications and corrections.
Reviewer 2 Report
I would like to thank the authors the effort to revise the manuscript. The interest on scaling up the production of nanoparticles warrants the publication of this report even as a proof-of-concept. Hence, in my opinion, the manuscript is now suitable for publication.